# New Challenges in Evaluating Outcomes after Immunotherapy in Recurrent and/or Metastatic Head and Neck Squamous Cell Carcinoma

**DOI:** 10.3390/vaccines10060885

**Published:** 2022-06-01

**Authors:** Andrea Alberti, Luigi Lorini, Marco Ravanelli, Francesco Perri, Marie Vinches, Paolo Rondi, Chiara Romani, Paolo Bossi

**Affiliations:** 1Medical Oncology Unit, Department of Medical & Surgical Specialties, Radiological Sciences & Public Health, ASST Spedali Civili di Brescia, University of Brescia, 25123 Brescia, Italy; a.alberti015@unibs.it (A.A.); l.lorini001@unibs.it (L.L.); 2Radiology Unit, Department of Medical & Surgical Specialties, Radiological Sciences & Public Health, ASST Spedali Civili di Brescia, University of Brescia, 25123 Brescia, Italy; marcoravanelli@hotmail.it (M.R.); p.rondi@unibs.it (P.R.); 3Medical and Experimental Head and Neck Oncology Unit, INT IRCCS Foundation G Pascale, 80131 Naples, Italy; f.perri@istitutotumori.na.it; 4Medical Oncology Department, Institut Régional du Cancer de Montpellier (ICM), 34090 Montpellier, France; marie.vinches@icm.unicancer.fr; 5Angelo Nocivelli Institute of Molecular Medicine, ASST Spedali Civili di Brescia, University of Brescia, 25123 Brescia, Italy; chiara.romani@unibs.it

**Keywords:** immunotherapy, response criteria, recurrent and/or metastatic head and neck squamous cell carcinoma, pseudoprogression, hyperprogression

## Abstract

In many recurrent and/or metastatic cancers, the advent of immunotherapy opens up new scenarios of treatment response, with new phenomena, such as pseudoprogression and hyperprogression. Because of this, different immune-related response criteria have been developed, and new therapeutic strategies adopted, such as treatment beyond progression. Moreover, the role of progression-free survival as a surrogate has been questioned, and new surrogate endpoint hypotheses have arisen. A proper understanding of radiological imaging, an assessment of the biological events triggered by therapy, and the clinical evolution of the lesions and of the patient performance status are all factors that should be considered to guide the oncologist’s treatment choice. The primary aim of this article is to discuss how all these concepts apply to recurrent/metastatic head and neck squamous cell carcinoma patients when treated with immunotherapy.

## 1. Introduction

Head and neck squamous cell carcinomas (HNSCCs) are heterogeneous tumors with a broad spectrum of prognoses, based mainly on the tumor stage. Within this group of malignancies, recurrent and/or metastatic (RM) HNSCCs are serious and life-threatening diseases, with a dismal prognosis [1]. The systemic treatment for RM HNSCC, in the last 10 years, has been mainly based on platinum chemotherapy plus either 5 fluorouracil or taxane and cetuximab, which are associated with high rates of toxicities [2,3]. The advent of first- and second-line immune checkpoint inhibitors (ICIs) has revolutionized treatment in this disease, hence providing unprecedented long-term survival and unexpected responses to treatment. CheckMate 141 and KEYNOTE 040 clinical trials demonstrated a gain in overall survival (OS) in patients with platinum-resistant RM HNSCCs treated with nivolumab and pembrolizumab over the investigator’s choice of chemotherapy (hazards ratio [HR]: 0.70; 95% CI: 0.51–0.96; *p* = 0.01; and HR: 0.80; 95% CI: 0.65–0.98; *p* = 0.0161, respectively) [4,5,6]. In platinum-naïve patients, the results from the KEYNOTE 048 study show a benefit in survival for pembrolizumab, either alone or in combination with platinum-based chemotherapy, over standard treatment [7]. The advent of immune checkpoint inhibitors (ICIs) for platinum-naïve RM HNSCC patients opens up new scenarios of treatment response. A proper understanding of radiological imaging, an assessment of the biological events triggered by therapy, and the clinical evolution of the lesions and of the patient performance status should be considered to guide the oncologist’s choice. The clinician must know how to address new types of radiological response, such as pseudoprogression, employ treatment beyond progression, consider hyperprogression, and interpret progression-free survival (PFS). In this regard, RM HNSCCs represent a challenge. Obtaining a clinical response, especially at the loco-regional relapse, is often critical to improve quality of life and to avoid risk of bleeding and airway obstruction. An accurate prediction of response is desirable, but the predictive value of PD-L1 expression and the tumor mutational burden, as clinical features, is still limited [8].

The primary aim of this article is to discuss the response patterns of RM HNSCCs to immunotherapy, and how to integrate them into the treatment choice.

## 2. Response Criteria for Cancer Immunotherapy

The tumor shrinkage in the target lesions, even a few weeks after initial administration, showed the activity of direct cytotoxic chemotherapy. Various studies have indicated that achieving a response after the initial cycles of chemotherapy is predictive of complete remission and improved survival [9,10]. With the increasing role of ICIs in oncology fields, new patterns of response have been observed, such as an increase in the size of target lesions due to inflammation or cystic colliquation, and, consequently, new concerns about the interpretation and characterization of the treatment activity [11]. In this way, several immune-related response criteria have been developed, and the main characteristics of these criteria are resumed in Table 1. The main new concepts with respect to the response evaluation criteria in the solid tumor guidelines (RECIST), version 1.1, which aim at a reliable measure for the outcomes of immune-based treatments, are that the appearance of new lesions does not imply progression, and that the lesion size increase must be confirmed after a defined period of time (from 4 to 12 weeks, depending on the criteria) [10]. However, in HNSCC, these criteria have been sparsely assessed, and there are no clear guidelines. It is of critical importance to early identify the progression of disease localized in the head and neck. For instance, the potential for airway obstruction, surgical resection, or radiotherapy to the site may alter the course of treatment. Unfortunately, common dimensional criteria, such as multiparametric MRI (e.g., tumor volume and apparent diffusion coefficient), are of limited help [12].

## 3. Implication of New Pattern of Response

### 3.1. Pseudoprogression

It has been observed that tumor response may occur after the initial size increase following immunotherapy. This phenomenon, called pseudoprogression, could be related to an immune response within the tumor and would not be related to the tumor cell growth and proliferation and, consequently, the true progression of the disease [17,18]. Pseudoprogression must be carefully evaluated because it may cause treatment discontinuation in the absence of a true therapy failure. On the contrary, it must be distinguished from true progression in order not to delay the treatment discontinuation of an inactive treatment. This phenomenon is rare among HNSCC patients, as witnessed by the only 1 patient out of 45 who had an atypical response, with an initial tumor flare, followed by a complete response in KEYNOTE 012, and by the only 2 patients out of 240 who experienced growth in the target lesions, followed by response, in CheckMate 141 [19,20,21]. A possible explanation of this minor rate is that this phenomenon is related to the immune infiltrate, which is poor in HNSCCs. Therefore, although progression after immunotherapy should not systematically cause immunotherapy interruption, tumor enlargement should be considered in most of the cases as an authentic progression because the increase in size in HNSCC may be dangerous due to anatomical features, as it can lead to immediate and life-threatening organ damage. Moreover, delaying the next line of therapy after immunotherapy might preclude the possibility of achieving clinical response. In this regard, emerging data show how chemotherapy given beyond progression to immunotherapy can give an unexpected good response [22].

### 3.2. Hyperprogression

It has been observed that some patients do not benefit from immunotherapy, but rather experience an acceleration in the tumor growth. Hyperprogression is defined as an increase in the tumor growth rate (TGR) after treatment initiation by a factor of 2 [23], and it is associated with a shorter PFS, but not with lower OS [24,25]. Some reports have described a high rate (29%) of HNSCC patients experiencing hyperprogression during immunotherapy treatment, which is detected as an acceleration of the tumor growth kinetics upon therapy with PD-1/PD-L1 inhibitors [24,26]. While clinicopathologic factors, such as tumor histology, tumor size at baseline, or previous chemotherapy, did not predict hyperprogression [25], a correlation was identified with the presence of regional recurrence, but not with local recurrence or distant metastasis. This could be explained by the alteration in the immune system of the lymph nodes, which could have changed their immune microenvironments after they were irradiated [26]. In the search for molecular predictive markers of hyperprogression, little progress has been made so far, as the molecular mechanisms that underlie this phenomenon are still poorly understood. The presence of immunosuppressive cells, such as Tregs or other myeloid-derived suppressor cells in tumor tissues, along with the lack of immunogenic tumor antigens (i.e., low mutational burden), have all been proposed as primary and adaptive biological mechanisms of resistance to ICIs, but their role in the context of this phenomenon needs to be further assessed [27]. Specific genomic alterations, such as EGFR and MDM2/4 gene amplification, have been associated with accelerated progression under immunotherapy in different studies involving multiple tumor types [28]. Several biomarkers have been studied as possible predictive factors of response to systemic therapy, and they could be useful also in distinguishing hyperprogression from pseudoprogression, even if there are few data on HNSCC. For instance, in melanoma and in lung cancer, the circulating-tumor-DNA (ctDNA) levels, which reflect the blood-based mutational burden and chromosomal instability of tumor cells, have shown rapid and dramatic decreases in patients with pseudoprogression, in contrast to patients with true progression, where they increased [29,30].

It is important, from a practical point of view, to distinguish naturally aggressive disease from hyperprogression. It is well known that RM HNSCCs, especially if progressing after first-line treatment, undergo accelerated growth due to several events, such as the dedifferentiation of tumor cells, and reduced immune control induced by drug treatment [26]. Further research is needed to identify the predictive factors of hyperprogression, given the paucity of data present in the literature.

Nowadays, neither biomarkers nor clinical predictive factors allow for distinguishing pseudoprogression from hyperprogression. In this regard, it is important to integrate radiological assessment with clinical evaluation. In the case of the worsening of clinical conditions or symptoms, such as pain, dyspnea, or dysphagia, one should immediately consider the change in treatment, without waiting for another radiological assessment.

### 3.3. Treatment beyond Progression

Data suggest that treatment beyond progression (TBP) after ICI therapy confers prolonged survival in selected patients with asymptomatic progression of disease [31]. Increased survival with TBP was also shown in patients with RM HNSCC by a post hoc analysis of CheckMate 141. Tumor-burden reduction was observed in 15 of 60 patients (25%) who underwent TBP with nivolumab; the median OS was 12.7 months for patients receiving TBP with nivolumab, and 7.7 months in the overall intent-to-treat population [21]. It is possible that patients continuing treatment are also the ones with the best performance status, which thus introduces a selection bias. At the moment, no criteria exist, except for performance status and clinical benefit, to guide clinician decisions on which patient should receive TBP, and which patient should change treatment.

### 3.4. Role of Progression Free Survival

The significance of PFS after immunotherapy is being discussed, and it seems to have a different relevance than in trials with chemotherapy. Tumor progression corresponds either to increased tumor burden or to the detection of new lesions, both events on cross-sectional imaging, and the tumor response refers to the reduced size of the tumors or tumors disappearing. The KEYNOTE 040 results show that, although no significant PFS increase was obtained, the OS in the intent-to-treat population was significantly increased after treatment with pembrolizumab, compared to chemotherapy treatment in platinum-resistant RM HNSCC (*p* = 0.016) [6]. Similarly, the CheckMate 141 study confirms this data with nivolumab in a platinum-resistant RM HNSCC study, where the OS was significantly longer with nivolumab compared to with standard therapy (HR for death: 0.70; 97.73% CI: 0.51–0.96; *p* = 0.01). Nevertheless, the median PFS did not statistically differ between the nivolumab arm and the standard-of-care arm (2.0 months versus 2.3 months, respectively) [5]. Even the KEYNOTE 048 did not show any gain in the PFS of pembrolizumab, either alone or in combination with chemotherapy, compared to cetuximab with chemotherapy, regardless of the CPS value [7]. Because of the prolonged effect on the tumor growth and on the chemotherapy efficacy administered beyond progression, the significance of PFS during immunotherapy should be redefined. Oligo-progression is not rare during treatment with immunotherapy, and especially in patients who have prolonged periods of partial response or stable disease. In these cases, an ablative local treatment, such as stereotactic radiotherapy, might play an important role, as reported in other cancer types [31,32].

Progression-free survival 2 (PFS-2), which is defined as the time from randomization to progression upon the first subsequent therapy, has been proposed as an alternative surrogate endpoint for OS and has been endorsed for use by the European Medicines Agency [33]. Woodford, R.G. et al. found that, across diverse tumors and therapies, the treatment effect on PFS-2 has a moderate correlation with OS [34]. A systematic prospective evaluation of PFS-2 as an endpoint, and its correlation with OS in HNSCC, is necessary before it can be embraced as a valid surrogate for OS.

## 4. New Imaging Frontiers: A Novel Antigone?

As Antigone, Oedipus’ daughter, guided her blind father in the famous Sophocles tragedy, recent branches of imaging are rising in order to guide older imaging techniques. New assessment methods aim at reading what happens inside the tumor before and during immunotherapy. Different techniques have been developed to improve the data sourcing from images, such as radiomics, or that aim at evaluating the metabolic performance, such as immunoPET, but they need to be validated for their ability to describe the tumor response to new treatments.

ImmunoPET is a promising noninvasive method to detect the CD8-dependent response to immunotherapy and the global PD-L1 expression within the tumor, and the assessment score should embed such information [35]. A number of response criteria have been proposed to overcome the limits of the assessment criteria, and they are currently under evaluation in some clinical trials (none of them are focused on head and neck neoplasms) [36]. A study with 89Zr-pembrolizumab PET imaging in melanoma and lung cancer shows that patients with high uptake showed a longer PFS and OS than patients with low uptake [37].

Another promising technique that is gaining interest, in both the clinical and radiological fields, is radiomics. Radiomics is the analysis of medical images by data-characterization algorithms, performed to obtain quantitative information that cannot be appreciated by the visual observation of the operator. Proper statistical methods (borrowed from biostatistics, applied to genomics) are used in this field to identify the more informative parameters in order to build predictive and/or prognostic models.

Promising results have been achieved in predicting HNSCC patient outcomes by using both CT- and MRI-based signatures [38,39]. When considering immunotherapy, few attempts have been made to predict the expression of PD-L1 [40,41] with good predictive efficacy. More should be undertaken to have a radiomic predictive model of the response to immune checkpoint inhibitors.

## 5. Conclusions

The advent of immune checkpoint inhibitors for platinum-naïve RM HNSCC patients opens up new scenarios of tumor response. Despite this, the traditional criteria for the assessment of chemotherapy outcomes remain the standard tools also for immunotherapy outcomes. After the advent of immunotherapy, clinicians have to face new phenomena, which add to the difficulties in the interpretation of the response to systemic therapies. At the first pseudoprogression, which should be properly differentiated from the true progression. This phenomenon is rare in HNSCC, and it is described in less than 1% of the patients enrolled in Checkmate 141 and Keynote 012. Another challenge introduced by immunotherapy is hyperprogression. Some reports have described a high rate (29%) of HNSCC patients experiencing this rapid progression of disease during immunotherapy [24,26]. Unfortunately, there are neither radiological factors nor biomarkers that may help identify patients at higher risk.

In conclusion, clinicians need to correctly interpret treatment responses to make correct decisions, but knowledge of biological mechanisms, and the validation of markers and imaging techniques, need further study. In RM HNSCC patients, a radiological progression may not necessarily be associated with treatment failure; on the other hand, careful follow up is needed to early detect possible hyperprogression. In the future, the proper integration of standard radiological imaging and new promising techniques, clinical evaluation, and circulating biomarkers will be the optimal way to define the response to immunotherapy and to make adequate treatment choices.

## Figures and Tables

**Table 1 vaccines-10-00885-t001:** Comparison between RECIST 1.1; irRECIST, iRECIST, imRECIST.

Characteristic	RECIST 1.1 [10]	irRC [13]	IR RECIST [14]	ImRECIST [15]	IRECIST [16]
PD	Increase of 20% in the sum of smallest diameter with an increase of at least 5 mm	>25% increase in SLD in two consecutive observations at least 4 weeks apart	IrPD: Increase of 20% TMTB (total measured tumor burden) compared with nadir or progression of nontarget lesions or new lesions	Increase > 20% in SLD compared with Nadir	IUPD: Increase > 20% of SLD (sum of longest diameter)
New Lesions	Progression disease	Results in PD that have to be confirmed in two observations at least 4 weeks apart	Longest diameter added to TMTB	Incorporated into total tumor burden	Not incorporated into total tumor burden
Confirmed PD	Not necessary	Required	Appearance of new lesions or unequivocal progression from initial IrPD after 4 weeks assessment	If after 4 weeks assessment, the evaluation is non-PD, the disease is updated to non-PD	After 4 weeks since first IUPD, becomes ICPD if increased size of target and nontarget lesion, increase in the sum of new target lesions > 5 mm, if appearance of another lesion.

Legends: irRC (immune-related response criteria); irRECIST (immune-related RECIST); imRECISR (immune-modified RECIST); iRECIST (immune RECIST); PD (progressive disease); RECIST (response evaluation criteria in the solid tumor) SLD (sum of the longest diameter); irPD (immune-related progressive disease); TMTB (total measured tumor burden); IUPD (immune-unconfirmed progressive disease); ICPD (immune-confirmed progressive disease).

## Data Availability

Not applicable.

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
