# Peer review of "New Challenges in Evaluating Outcomes after Immunotherapy in Recurrent and/or Metastatic Head and Neck Squamous Cell Carcinoma"

_vaccines, 2022, doi:10.3390/vaccines10060885_

Round 1
Reviewer 1 Report
The submitted manuscript entitled “New challenges in evaluating outcomes after immunotherapy in recurrent and/or metastatic head and neck squamous cell carcinoma” authored by Dr. Alberti and others is not recommended for publication based on my understanding. The paper provides very limited information and discussion, so I cannot give a full evaluation.
Author Response
The submitted manuscript entitled “New challenges in evaluating outcomes after immunotherapy in recurrent and/or metastatic head and neck squamous cell carcinoma” authored by Dr. Alberti and others is
not recommended for publication based on my understanding. The paper provides very limited information and discussion, so I cannot give a full evaluation.
Reply: A daily challenge for clinicians who treat patients with head and neck cancer is to evaluate the outcomes after immunotherapy. For these clinicians be update on new phenomena like pseudoprogression and hyper-progression or like the influence of immunotherapy on the PFS2, is important, especially because these aspects are not so defined. To detect early the progression of a disease that infiltrate deep tissue in the neck region could affect the prognosis. The aim of this paper is to resume knowledge on this topic, and provide suggestions to clinicians that face with these it every day.
Reviewer 2 Report
The authors have presented a paper about "New challenges in evaluating outcomes after immunotherapy in recurrent and/or metastatic head and neck squamous cell carcinoma".
The topic itself is absolutely interesting however I feel that some additions would much improve the level of the manuscript.
1) The target of recurrence and metastases should be more clearly defined: do the authors focus on lymph nodes metastates or on distant sites? please clarifiy because one thing is focusing on the neck region imaging and another thing is extending to the thorax and over.
2) Which are currently the clinical and/or radiological factors which allow to discriminate between psudoprogression and hyperprogression? Please try to suggest some proposal for clinical purposes based on available literature.
3) It would be interesting to investigate the previous local treatment strategy in cases of recurrence (surgery? radiotherapy?): are there any differences in the response to immunotherapy based on different local different approaches?
4) It is more and more a matter of debate in the scientific community the combination of immotherapy and local therapies to increase immunotherapy results by exposing more antigens. With this perspective radiotherapy plays a major role, also in the case of reirradiation (see PMID 28291904 for further details).
5) Please explore the possible contribution (if any) of radiomics with regard to the topic of the article
Author Response
1) The target of recurrence and metastases should be more clearly defined: do the authors focus on lymph nodes metastates or on distant sites? please clarifiy because one thing is focusing on the neck region imaging and another thing is extending to the thorax and over.
Reply: Thanks for this suggestion. We have clarified this aspect in the article. We don’t want to focalize the attention on only one of these aspects, even if we dedicated more room to head and neck area.
2) Which are currently the clinical and/or radiological factors which allow to discriminate between psudoprogression and hyperprogression? Please try to suggest some proposal for clinical purposes based on available literature.
Reply: We have improved the suggestions on this aspect, according to data present in literature.
3) It would be interesting to investigate the previous local treatment strategy in cases of recurrence (surgery? radiotherapy?): are there any differences in the response to immunotherapy based on different local different approaches?
Reply: Unfortunately, we have found only few data about this question. We have integrated them into the paper.
4) It is more and more a matter of debate in the scientific community the combination of immunotherapy and local therapies to increase immunotherapy results by exposing more antigens. With this perspective radiotherapy plays a major role, also in the case of reirradiation (see PMID 28291904 for further details).
Reply: We have implemented this topic into the article.
5) Please explore the possible contribution (if any) of radiomics with regard to the topic of the article
Reply: We have asked to a radiology expert to improve this aspect. See the changes introduced in the text in this regard.
Reviewer 3 Report
REVIEW
In this Review the Authors focused on the new outcomes of recurrent and/or metastatic cancer immunotherapy treatment and new scenarios of responses like pseudoprogression and hyperprogression. Also, the Authors discussed different immune-related response criteria and to new therapeutic strategies adopted, like treatment beyond progression. Particular attention was payed to the factors that should be considered to guide the oncologist’s treatment choice, including the appropriate analysis of radiological imaging, assessment of biological events triggered by therapy, and clinical evolution of the lesions and of patient performance status.
This is an important review providing deep analysis of current immune-related response criteria of cancer treatment, new therapeutic strategies and all factors that should be considered to guide the oncologist’s treatment choice. This article can be published. There are several small concerns.
- Please remove excessive abbreviation where it is possible.
For example, you can easily remove abbreviations
(RM) and (HNSCC) from the Abstract.
- At the same time, please, unfold abbreviation RECIST 1.1. at line 65 (Chapter 2, page 2). RECIST - Response Evaluation Criteria in Solid Tumours guidelines, version 1.1.
- At line 63, Chapter 2, Chapter 2 please insert parenthesis near the ref 10 ([10])
- Table is quite difficult for perception
If it is possible, please, remove excessive abbreviation or, visually appealing figure
- Please, re-write Conclusion.
It would be good to redirect logic flow from difficulties, knowledge gaps and inconsistencies towards perspectives and key points of the Review.
Author Response
- Please remove excessive abbreviation where it is possible.
- At the same time, please, unfold abbreviation RECIST 1.1. at line 65 (Chapter 2, page 2). RECIST - Response Evaluation Criteria in Solid Tumours guidelines, version 1.1.
- At line 63, Chapter 2, Chapter 2 please insert parenthesis near the ref 10 ([10])
Reply: We have met your requirements
- Table is quite difficult for perception
Reply: As you suggest we have improved the table perception
- Please, re-write Conclusion
Reply: We followed your advice and rewrote the conclusions.
Round 2
Reviewer 2 Report
In the present form the article still has serious flaws. Please seriously consider my previous comments.
Author Response
We tried to further implement the paper according to your suggestions, but without going outside the topic of the article "evaluating outcomes after immunotherapy".
